# Soil Fungal Diversity and Ecology Assessed Using DNA Metabarcoding along a Deglaciated Chronosequence at Clearwater Mesa, James Ross Island, Antarctic Peninsula

**DOI:** 10.3390/biology12020275

**Published:** 2023-02-09

**Authors:** Vivian N. Gonçalves, Juan M. Lirio, Silvia H. Coria, Fabyano A. C. Lopes, Peter Convey, Fábio S. de Oliveira, Micheline Carvalho-Silva, Paulo E. A. S. Câmara, Luiz H. Rosa

**Affiliations:** 1Departamento de Microbiologia, Universidade Federal de Minas Gerais, Belo Horizonte 31270-901, Brazil; 2Instituto Antártico Argentino, 25 de Mayo 1143, General San Martín, Buenos Aires B1650HMK, Argentina; 3Laboratório de Microbiologia, Universidade Federal Do Tocantins, Porto Nacional 77500-000, Brazil; 4British Antarctic Survey, NERC, High Cross, Madingley Road, Cambridge CB3 0ET, UK; 5Department of Zoology, University of Johannesburg, P.O. Box 524, Auckland Park 2006, South Africa; 6Millennium Institute Biodiversity of Antarctic and Subantarctic Ecosystems, BASE, University Austral of Chile, Valdivia 5090000, Chile; 7Cape Horn International Center (CHIC), O’Higgins 310, Puerto Williams 6350000, Chile; 8Departamento de Geografia, Universidade Federal de Minas Gerais, Belo Horizonte 31270-901, Brazil; 9Departamento de Botânica, Universidade de Brasília, Brasília 70297-400, Brazil; 10Laboratório de Microbiologia Polar e Conexões Tropicais, Instituto de Ciências Biológicas, Universida-de Federal de Minas Gerais, Belo Horizonte 31270-901, Brazil

**Keywords:** Antarctica, extremophiles, environmental DNA, fungi, metabarcoding, taxonomy

## Abstract

**Simple Summary:**

We characterized the fungal organisms present in soils at James Ross Island, north-east Antarctic Peninsula, using DNA metabarcoding. Taxa detected included members of the widespread phyla *Ascomycota*, *Basidiomycota* and *Mortierellomycota*. Additionally, the uncommon phyla *Chytridiomycota*, *Rozellomycota*, *Monoblepharomycota*, *Zoopagomycota* and *Basidiobolomycota* were detected. Unknown fungi and taxa identified at generic and specific levels dominated the assemblages. The fungal sequence assemblages displayed high diversity and richness, and moderate dominance, and included taxa known to play saprophytic, pathogenic and symbiotic functions. Soils of Clearwater Mesa contain a complex fungal community, including fungal groups considered rare in Antarctica, dominated by cold-adapted cosmopolitan, endemic, saprotrophic and phytopathogenic taxa.

**Abstract:**

We studied the fungal diversity present in soils sampled along a deglaciated chronosequence from para- to periglacial conditions on James Ross Island, north-east Antarctic Peninsula, using DNA metabarcoding. A total of 88 amplicon sequence variants (ASVs) were detected, dominated by the phyla *Ascomycota*, *Basidiomycota* and *Mortierellomycota*. The uncommon phyla *Chytridiomycota*, *Rozellomycota*, *Monoblepharomycota*, *Zoopagomycota* and *Basidiobolomycota* were detected. Unknown fungi identified at higher hierarchical taxonomic levels (Fungal sp. 1, Fungal sp. 2, *Spizellomycetales* sp. and *Rozellomycotina* sp.) and taxa identified at generic and specific levels (*Mortierella* sp., *Pseudogymnoascus* sp., *Mortierella alpina*, *M. turficola*, *Neoascochyta paspali*, *Penicillium* sp. and *Betamyces* sp.) dominated the assemblages. In general, the assemblages displayed high diversity and richness, and moderate dominance. Only 12 of the fungal ASVs were detected in all chronosequence soils sampled. Sequences representing saprophytic, pathogenic and symbiotic fungi were detected. Based on the sequence diversity obtained, Clearwater Mesa soils contain a complex fungal community, including the presence of fungal groups generally considered rare in Antarctica, with dominant taxa recognized as cold-adapted cosmopolitan, endemic, saprotrophic and phytopathogenic fungi. Clearwater Mesa ecosystems are impacted by the effects of regional climatic changes, and may provide a natural observatory to understand climate change effects over time.

## 1. Introduction

Antarctica hosts largely undisturbed ecosystems, many of which have been the subject of limited biological research. These ecosystems and their biodiversity are exposed to multiple extreme conditions that combine cold, dry and ultra-oligotrophic stresses, offering unique opportunities to discover and study extremophilic organisms [1,2]. Typically, Antarctic soils are dry, and are exposed to large and rapid changes in temperature, and high levels of solar radiation. They are also generally nutrient-poor, typically ranging between moderately and ultra-oligotrophic [3].

Clearwater Mesa, located on James Ross Island, includes approximately 8 km^2^ of ice-free area of volcanic geology, and lies at up to 250 m a.s.l. on the south-east side of Croft Bay [4]. Clearwater Mesa is part of the James Ross Island Volcanic Group and comprises rocks of the Pliocene age, composed mainly of alkaline basalts and palagonitized hyaloclastite breccias, covered by glacial deposits with basaltic clasts [5]. In ecology, chronosequence soils represent a set of sites formed from the same substrate that differ in age and the levels of geological and biological development since they were formed [6]. Clearwater Mesa has a characteristic chronosequence profile from the seashore to the Haddington Ice Cap, which encompasses a gradient from para- to periglacial conditions.

The soils of Antarctica offer opportunities to investigate the regional-to-global environmental effects of microbiota on community structure and function [7]. The Antarctic Peninsula region has been strongly influenced by the effects of regional climate change since the mid-20th Century [8] and, therefore, provides an attractive field laboratory for the study of different aspects of fungal biology under extreme conditions [2,9]. Most studies to date of Antarctic fungal diversity have focused on soils [3,10]. However, the majority of these studies relied on traditional culturing approaches [11], which, as is typical of microbiota, do not reveal the full microbial diversity present, and particularly, that of basal, cryptic and rare phyla. The recent availability and application of molecular biological culture-independent approaches have highlighted that fungal diversity in Antarctic soils is likely to be considerably greater than previously appreciated [9,11,12]. In the current study, we applied a DNA metabarcoding approach using high throughput sequencing (HTS) to detect and characterize the soil fungal community along a chronosequence from para- to periglacial conditions at Clearwater Mesa, James Ross Island.

## 2. Methods

### 2.1. Study Sites and Sediment Sampling

Four topsoil samples were collected in the Clearwater Mesa region of James Ross Island, north-east Antarctic Peninsula, across an environmental gradient or chronosequence from para- (Site 1) to periglacial (Sites 2, 3 and 4) conditions (Figure 1). The characteristics of each sampling point are presented in Table 1. Topsoils were collected during the austral summer in February 2021 (Appendix A). The chronosequence extended 4770 m, from the coast to the edge of the Haddington Ice Cap, with an approximately NW-SE (azimuth 114°) orientation. All samples were collected over the volcanic bedrock of basaltic composition belonging to the James Ross Island Volcanic Group. Sampling locations were selected to be free of lichens and mosses. All soil samples were collected from the center of sorted stone circles, comprising mainly clay material. Soil samples were collected in triplicate and immediately placed in sterile Whirl-pack bags (Nasco, Ft. Atkinson, WI, USA). Approximately 500 g of each topsoil sample was then immediately sub-sampled (in triplicate), sealed, placed in sterile Whirl-pack bags and frozen at −20 °C until processing in the laboratory at the Federal University of Minas Gerais, Brazil. There, the samples were gradually thawed at 4 °C for 24 h before carrying out DNA extraction.

### 2.2. Soil Physical and Chemical Analyses

The granulometric soil characteristics of sand, silt and clay soil chemical analyses were determined following Teixeira et al. [13]. All analyses were performed in triplicate.

### 2.3. DNA Extraction, Illumina Library Construction and Sequencing

Three replicate sub-samples were obtained from each soil sample under strict control conditions to avoid external contamination, as described by Rosa et al. [11]. Total DNA was extracted from these using the FastDNA Spin Kit for Soil (MPBIO, Solon, OH, USA). DNA quality was analyzed using agarose gel electrophoresis (1% agarose in 1× Trisborate-EDTA) and then quantified using the Quanti-iT™ Pico Green dsDNA Assay (Invitrogen). The internal transcribed spacer 2 (ITS2) of the rDNA was used as a DNA barcode for fungal identification [14,15], using the primers ITS3 and ITS4 [16]. The Herculase II Fusion DNA Polymerase Nextera XT Index Kit V2 was used for library construction and DNA amplification followed the Illumina 16S Metagenomic Sequencing Library Preparation protocol (Part #15044223, Rev. B). Paired-end sequencing (2 × 300 bp) was performed on a MiSeq platform (Illumina) by Macrogen Inc. (Seoul, South Korea).

### 2.4. Data Analysis and Fungal Identification

Quality analysis of the libraries was performed using BBDuk v. 38.87 in BBmap software, as described by Bushnell [17], with the following parameters: Illumina adapters removing (Illumina artefacts and the PhiX Control v3 Library); ktrim = l; k = 23; mink = 11; hdist =1; minlen = 50; tpe; tbo; qtrim = rl; trimq = 20; ftm = 5; maq = 20. The remaining sequences were imported to the QIIME2 version 2021.4 (https://qiime2.org/ (accessed on 1 July 2022)) for bioinformatics analyses [18]. The qiime2-dada2 plugin was applied for filtering, dereplication, turn paired-end fastq files into merged and chimera removal using default parameters [19]. Taxonomic assignments were determined for amplicon sequence variants (ASVs) in the steps: (1) ASVs were classified using the qiime2-feature-classifier [20] classify-sklearn against the UNITE Eukaryotes ITS database version 8.3 [21]; (2) Remaining unclassified ASVs were filtered and aligned against the filtered NCBI non-redundant nucleotide sequences (nt) database (October 2021) using BLASTn [22] with default parameters; the nt database was filtered using the following keywords: “ITS1″, “ITS2″, “Internal transcribed spacer″ and “internal transcribed spacer″; and (3) Output files from BLASTn [22] were imported to MEGAN6 [23] and taxonomic assignments were performed using the “megan-nucl-Jan2021.db″ mapping file with default parameters and trained with Naive Bayes classifier and a confidence threshold of 98.5%. Krona [24] was used for generating taxonomic profiles. The heatmap of ASV abundance and clustering analysis were produced using Heatmapper [25] following the parameters: average linkage, spearman rank correlation and Z-score, among samples for each ASV.

According to Medinger et al. [26], different factors, such as extraction, PCR and primer bias can affect the number of DNA reads obtained, and for this reason, lead to the misinterpretation of absolute DNA read abundance [27]. However, according to Giner et al. [28], such biases do not affect the proportionality between reads and cell abundance, implying that more reads are linked with higher abundance [29,30]. For comparative purposes, we used the number of DNA reads as a proxy for relative abundance. For fungal classification we followed Kirk et al. [31], Tedersoo et al. [32] and MycoBank (http://www.mycobank.org (accessed on 1 July 2022)).

### 2.5. Fungal Diversity and Ecology

The relative abundance (RA) criteria established by Rosa et al. [33] were used, where fungal ASVs > 10% were considered dominant, those between 1% and 10% as intermediate and those with <1% as minor components of the fungal community. Relative abundance values were used to quantify fungal assemblage diversity, richness, dominance and similarities using the Fisher’s α, Margalef’s, Simpson’s, Sorensen, Bray–Curtis and Mao Tao indices. All indices were obtained with 95% confidence, with bootstrap values calculated from 1000 replicates by the PAST program 1.90 [34]. Ecological functional assignments of fungal taxa’s generic levels were obtained using FunGuild [35].

## 3. Results

### 3.1. Fungal Taxonomy

A total of 127,774 DNA reads were detected in the four soils sampled, which represented 88 fungal ASVs (Appendix A). The Mao Tao curves of the assemblages present in Sites 2 and 4 reached the asymptote. However, those of Sites 1 and 3 did not reach asymptote, indicating that further fungal sequence diversity is likely to be present in these soils (Appendix A). The relative abundances of representative fungi at different hierarchical levels varied across the four sites (Appendix A). The metabarcoding approach revealed the presence of taxa of fungal phyla often detected in Antarctica by culturing and eDNA methods (*Ascomycota*, *Basidiomycota* and *Mortierellomycota*). However, metabarcoding also detected the more rarely detected fungal phyla *Chytridiomycota*, *Rozellomycota*, *Monoblepharomycota*, *Zoopagomycota* and *Basidiobolomycota*.

Unknown fungi identified at higher hierarchical taxonomic levels (Fungal sp. 1, Fungal sp. 2, *Spizellomycetales* sp. and *Rozellomycotina* sp.), as well as taxa identified at generic and specific levels (*Mortierella* sp., *Pseudogymnoascus* sp., *Mortierella alpina*, *M. turficola*, *Neoascochyta paspali*, *Penicillium* sp. and *Betamyces* sp.) dominated the fungal assemblages. However, the majority of diversity present in the assemblages represented fungal taxa detected at intermediate and rare relative abundance. A total of 35 fungal ASVs (39.7%) could only be assigned to phylum, class, order or family (higher taxonomic levels).

### 3.2. Geological and Physicochemical Characteristics of Soil Sampling Sites in Relation to Fungal Assemblages Detected

The soils showed homogeneous granulometric composition and a loam texture in the paraglacial zone (Site 1), transitioning to clay-sand-loam in the periglacial zone (Sites 2, 3 and 4). The contents of fine particles (clay and silt) were similar in all soils (approximately 50%), as were the contents of coarse and fine sand. Additionally, there was a homogeneous distribution of fractions, regardless of the parent material of the soils. The soils had alkaline pH, always above 7 and reaching 8.9. All soils were eutrophic, with high base saturation, reaching 100%, with zero potential acidity. Only the soil from Site 2 showed slightly lower values of CEC and sum of bases. The organic carbon content was very low (<1%) and the P content was distinct only in the soil from Site 1.

The diversity indices of the fungal assemblages detected varied across the different sampling sites but, in general, they displayed high diversity and richness, and moderate dominance (Table 1). The soil sampled from Site 2 displayed the highest number of assigned taxa and diversity indices, followed by those of Sites 3, 4 and 1. Geologically, the soils at each site were located over a similar bedrock. Site 4 was located only 50 m from the retreating Haddington Ice Cap front, which we assume represents a very young ice-free area. The location of Site 1 is closest to the coast of Croft Bay and may be more exposed to sea spray, although is still at about 250 m a.s.l. on the mesa. There was no indication of any link between the geological or physicochemical parameters measured and the diversity indices obtained.

### 3.3. Fungal Assemblage Distribution along the Soil Chronosequence

The Sorensen and Bray–Curtis indices were applied to compare the fungal assemblage distribution along the chronosequence soils (Appendix A). Both similarity indices showed the same pattern, with the assemblages detected at Site 2 differing from those of Sites 1, 3 and 4. Of the 88 fungal ASVs, only 12 (13.6%) were detected in all soil samples (Figure 2; Appendix A). Fungal ASV distribution changed across the sites, with the soils sharing some taxa and also hosting some specific fungal taxa. Site 2 hosted 29 exclusive ASVs. The most abundant fungal ASVs, Fungal sp. 1, Fungal sp. 2, *Betamyces* sp., *Penicillium* sp., *Pseudogymnoascus* sp., *Spizellomycetales* sp. and *Neoascochyta paspali* were present at all four sites sampled. Functional ecology assignments of the fungal ASVs at the generic level (Appendix A) suggested that the assemblages detected in the soils were dominated by saprophytic, plant and animal pathogenic and symbiotic taxa.

## 4. Discussion

### 4.1. Fungal Taxonomy

Soils represent perhaps the most studied substrate as habitat for fungi in Antarctica [2]. Until recently, most fungal biodiversity studies have used traditional culture-dependent methods [10]. However, culture-independent methodologies have increasingly become available in recent years, and have started to be applied to Antarctic soils. For instance, Baeza et al. [12] applied a culture-independent approach to characterize fungi in soils from different Antarctic regions, reporting taxa of the phyla *Ascomycota*, *Basidomycota*, *Chytridiomycota* and *Neocallimastigomycota*. Newsham et al. [9] reported taxa of the phyla *Ascomycota*, *Basidiomycota*, *Chytridiomycota*, *Glomeromycota* and *Mucoromycota* in soils sampled from maritime Antarctica. Rosa et al. [11] similarly reported DNA sequences of taxa belonging the phyla *Ascomycota*, *Basidiomycota*, *Mortierellomycota*, *Mucoromycota*, *Chytridiomycota* and *Rozellomycota* in soils subject to and protected from human influence on Deception Island. In common with these studies, we detected taxa of the phyla *Ascomycota*, *Basidiomycota*, *Mortierellomycota*, *Chytridiomycota* and *Rozellomycota*. However, we also detected the DNA of taxa belonging to the more rarely reported phyla *Monoblepharomycota*, *Zoopagomycota* and *Basidiobolomycota*. In agreement with the studies of Newsham et al. [9], Rosa et al. [11] and Baeza et al. [12] we detected 35 ASVs (39.7% of the total assigned) in the sampled soils that could only be classified to higher taxonomic levels, suggesting that it is likely that Antarctica hosts many as yet unrecognized fungal taxa or taxa not currently included in the consulted sequence databases. Fungal sp. 1, Fungal sp. 2, *Spizellomycetales* sp., *Rozellomycotina* sp., *Mortierella* sp., *Pseudogymnoascus* sp., *Mortierella alpina*, *M. turficola*, *Neoascochyta paspali*, *Penicillium* sp. and *Betamyces* sp. were the most abundant fungal taxa detected in the Clearwater Mesa soils. However, we recognize that, as our study focused on fungal diversity assigned from DNA sequences, further specific studies are necessary to elucidate whether these fungi are present in viable form.

The phyla *Chytridiomycota* and *Rozellomycota* include freshwater zoosporic fungi rarely reported in Antarctic environments [11,33]. Members of both phyla have been reported from ice-covered lakes in the McMurdo Dry Valleys [36]. Some representatives of these phyla are parasites of other taxa, including fungi, algae and aquatic invertebrates [37,38]. *Spizellomycetales* (an order of *Chytridiomycota*) includes 27 species, including cosmopolitan and frequently found saprophytes and/or parasites in freshwater and soil [31]. *Spizellomycetales* species, such as members of the genera *Spizellomyces* and *Gaertneriomyces*, are considered important plant pathogens [39]. The genus *Betamyces* (*Chytridiomycota*) currently includes only a single recognized species, *Betamyces americae-meridionalis*, reported from pollen baits at the Paraná River, Buenos Aires, Argentina [40]. *Betamyces* spp. have also been reported from soil in Costa Rica [40]. Recently, using the same metabarcoding approach as in the current study, Gonçalves et al. [4] reported *Spizellomycetales* sp., *Rozellomycota* sp. and *Betamyces* sp. as dominant fungal taxa present in lake sediments that were also sampled from Clearwater Mesa.

*Pseudogymnoascus* and *Mortierella* are genera that include various psychrophilic species, with many commonly present in different Antarctic environments. *Pseudogymnoascus* (syn. *Geomyces*) and *Mortierella* are commonly reported from cold habitats of Arctic, alpine, Antarctic and temperate ecosystems [2]. In Antarctica, both genera are widely distributed in terrestrial and marine ecosystems, including soils, mosses, lichens, plants, macroalgae, lake and marine sediments and soils [41]. Members of the genus *Penicillium* occur globally, including some truly cosmopolitan species, and have been reported from multiple substrates in Antarctica, including soils [10,42,43] and permafrost [44,45], and are associated with macroalgae [3]. *Pseudogymnoascus*, *Mortierella* and *Penicillium* taxa have been detected in metabarcoding studies of soils and other Antarctic substrates such as rocks [46], marine sediments [47], mosses [41], snow [48] and lake sediments [49,50]. The genus *Neoascochyta* includes 18 recognized species [51], including the phytopathogenic *N. paspali* [52]. In Antarctica, *N. paspali* (syn. *Phoma paspali*) was detected in a study based on culture-dependent methods from a freshwater lake on King George Island [53]. DNA sequences of *N. paspali* have also been reported in air and snow samples from Livingston Island [54] and in ice [46].

### 4.2. Fungal Distribution along the Chronosequence

Our sequence data revealed the potential presence of moderate-to-rich fungal diversity in all of the soils collected, but did not indicate any relationship between the diversity values obtained and the position of the soils within the chronosequence. The fungal sequence assemblage detected in soil sampled at Site 2 displayed the highest diversity indices and appeared distinct from those of Site 1, 3 and 4. However, no differences were identified in the physical (granulometry) and chemical parameters of the four sites. Only Site 2 soil showed lower exchangeable Ca values although, even so, the exchange complex remained saturated with bases and the soils were eutrophic. Further studies are necessary to clarify whether the lower Ca values at Site 2 have any influence on nutrient utilization in these soils, or a relationship with the greater fungal diversity recorded at this site.

Despite the lack of difference in soil properties between the four sites sampled here, there were some interesting features of the fungal ASV distribution along the chronosequence. *Pseudogymnoascus* sp., *Thelebolus balaustiformis* and *Antarctomyces* sp., all considered Antarctic endemics, displayed high or moderate abundance at Sites 2, 3 and 4, which have been more recently exposed by glacial retreat; in contrast, these taxa displayed their lowest abundance at Site 1, which may represent a more developed soil. In addition, Site 2, relatively distant from both the coast and glacier margin, generated the most DNA reads, which might indicate a warmer microclimate in comparison with the other sampling sites.

Comparing the fungal sequence diversity detected in these soils with those detected in lake sediments from Clearwater Mesa reported by Gonçalves et al. [4] and de Souza et al. [49] (Appendix A), the diversity and richness indices of the lake sediments were generally higher than those of the soils. This may be a result of the sediments accumulating different inputs over time (soil, organic matter and biological propagules) and, consequently, accumulating DNA of different organisms. Around 36% of fungal ASVs were shared between the soils and lake sediments of Clearwater Mesa, also consistent with a portion of the fungal DNA present in lake sediments originating from the surrounding soils.

### 4.3. Ecology

The fungal genera identified in the soils of Clearwater Mesa are characterized by different ecological functions, including saprophytes, mutualists, symbionts and parasites, as also reported in a study of soil from Deception Island [11], and from lake sediments on Clearwater Mesa [4,49]. Among the genera reported in Appendix A, saprophytes dominated the assemblages in all four sampling sites, followed by plant and animal pathogens and symbionts, a pattern consistent with multiple recent fungal metabarcoding studies in different Antarctic habitats [11,33,46,48,49,50,54,55,56]. Fungi are able to colonize polar environments and degrade organic matter at low temperatures, generating breakdown products containing carbon, nitrogen and other elements, which become available to other organisms [57]. Taken together, this body of research provides support for the hypothesis that Antarctic ecosystems host complex fungal communities that play vital roles in the decomposition of organic matter under extreme conditions.

## 5. Conclusions

The fungal sequence assemblages detected in this DNA metabarcoding study support the presence of a complex soil fungal community, including the presence of fungal groups generally considered rare in Antarctica (*Chytridiomycota*, *Rozellomycota*, *Monoblepharomycota*, *Zoopagomycota* and *Basidiobolomycota*). Soils collected along a chronosequence from para- to periglacial conditions hosted a fungal community dominated by cold-adapted cosmopolitan (*Penicillium*) and some endemic (*Mortierella*, *Pseudogymnoascus*, *Thelebolus* and *Antarctomyces*) taxa, including saprophytic (*Penicillium* and *Betamyces*) and phytopathogenic (*Neoascochyta*) representatives. The community was dominated, overall, by saprophytes, which are likely to contribute to organic matter decomposition in soils, thereby supplying other elements of the soil food web. Clearwater Mesa is in a region of the Antarctic Peninsula that is impacted by the effects of regional climatic changes and may provide a natural observatory for the study of climate change effects over time. 

## Figures and Tables

**Figure 1 biology-12-00275-f001:**
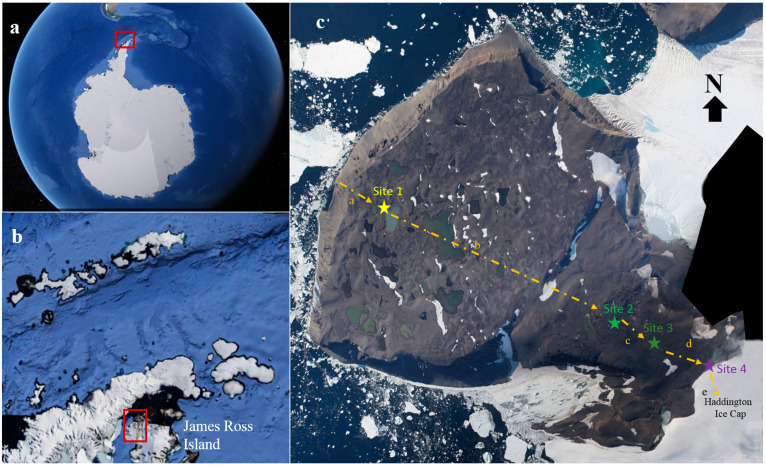
Satellite images (**a**,**b**) (obtained in Google Earth Pro, 2019). (**a**) Antarctica with the northern Antarctic Peninsula inside the red rectangle; (**b**) The western part of James Ross Island inside the red rectangle; (**c**) Clearwater Mesa with colored stars representing Site 1 (64°01′16.5″ S; 57°43′49.6″ W), site 2 (64°01′53.5″ S; 57°40′47.0″ W), Site 3 (64°02′03.6″ S; 57°40′05.9″ W) and Site 4 (64°02′09.5″ S; 57°39′22.1″ W) on James Ross Island. Photo **c** taken by Natacha Lopez. Yellow line represents the distances between the sites sampled: **a** = 650 m, **b** = 2700 m, **c** = 640 m, **d** = 730 m and **e** = 50 m.

**Figure 2 biology-12-00275-f002:**
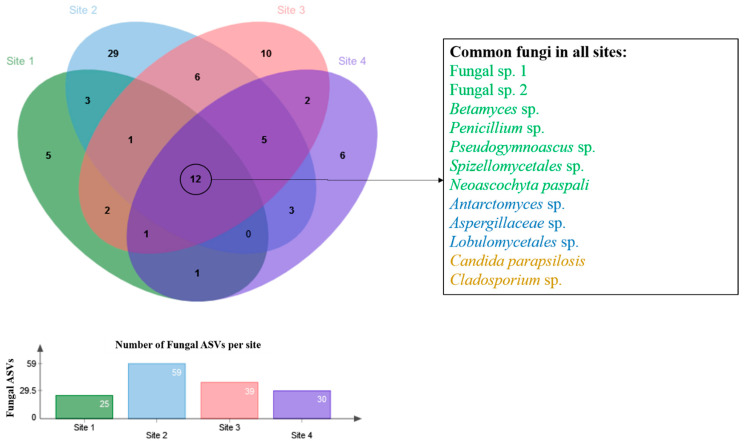
Venn diagram showing the distribution of fungal amplicon sequence variants (ASVs) across the four soil samples obtained at Clearwater Mesa, James Ross Island, north-east Antarctic Peninsula. In the key: green represents the dominant taxa, blue those of intermediate abundance and orange the rare taxa.

**Table 1 biology-12-00275-t001:** Locations, characteristics, soils physicochemical data and diversity indices of the fungal assemblages present in soil samples obtained at Clearwater Mesa, James Ross Island, north-east Antarctic Peninsula.

	Clearwater Mesa Soils			
Parameters	Site 1	Site 2	Site 3	Site 4
Geology				
Location	64°01′16.5″ S; 57°43′49.6″ W	64°01′53.5″ S; 57°40′47.0″ W	64°02′03.6″ S; 57°40′05.9″ W	64°02′09.5″ S; 57°39′22.1″ W
Altitude (meters above sea level)	247	182	186	201
Distance to coastline (m)	650	3350	3990	4720
Sampling site characteristics	Located in patterned ground over basaltic lava flow	Located in patterned ground over basaltic hyaloclastite breccias	Located in elongated patterned ground, very rich in clay sediment, formed over basaltic hyaloclastite breccias	Located in basal till deposit composed mainly by basaltic angular clast, with incipient formation of patterned ground, over basaltic hyaloclastite breccias
Sediment physical parameters				
Clay (%)	25	21	14	24
Silt	26	31	37	20
Coarse sand	25	27	21	28
Fine sand	24	20	28	22
Textural class	Clay-sand-loam	Loam	Loam	Loam
Sediment chemical parameters				
pH in H_2_O	8.8	8.3	8.9	7.6
Exchangeable P—mg dm^3^	103.2	13.6	6.7	10.7
Sum of exchangeable bases Ca+K+Mg (SB)—cmol_c_ dm^3^	8.81	4.54	6.54	8.75
Percentage of base saturation (PBS)-%	100.0	100.0	100.0	89.8
H+Al—potential acidity—cmol_c_ dm^3^	0.00	0.00	0.00	0.99
Cation exchange capacity at pH 7 (CEC)—cmol_c_ dm^3^	8.81	4.54	6.54	9.74
Total organic carbon (TOC)—dag kg^−1^	0.7	0.7	0.93	0.54
Micronutrient Fe—mg dm^3-1^	214.2	14.3	41.0	14.8
Micronutrient Mn—mg dm^3-1^	72.1	27.4	58.7	24.1
Fungal diversity indices				
Number of DNA reads	15,636	56,473	38,575	17,090
Number of taxa assigned	25	59	39	31
Fisher’s-α (diversity)	10.7	60.43	23.51	15.39
Margalef (richness)	5.21	12.6	8.25	6.51
Simpson’s (dominance)	0.81	0.83	0.84	0.86

## Data Availability

All raw sequences have been deposited in the NCBI database under the codes SAMN32942657, SAMN32942658, SAMN32942659, SAMN32942660, SAMN32942661, SAMN32942662, SAMN32942663, SAMN32942664, SAMN32942665, SAMN32942667 and SAMN32942668.

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
