# Peer review of "Soil Fungal Diversity and Ecology Assessed Using DNA Metabarcoding along a Deglaciated Chronosequence at Clearwater Mesa, James Ross Island, Antarctic Peninsula"

_biology, 2023, doi:10.3390/biology12020275_

Round 1

Reviewer 1 Report

Notes for

 Fungal and fungal-like diversity and ecology assessed using DNA metabarcoding along a deglaciated chronosequence soils at Clearwater Mesa, James Ross Island, Antarctic Peninsula

The MS in this form need major revision.

Here there are some corrections and considerations 

Simple summary

line

30 “Straminopila” change in “Straminipila” (see Mycobank or other atlas such as Fungal Biodiversity by Crous et al., 2019, Westerdijk Fungal Biodiversity Institute. Change also on line 43 and in all the text, if the authors want to report data on Straminipila. In any case Bacillariophyta contains diatoms and it is better to take it off in the text, all the table, figures and supplementary materials. Moreover it seems that only Bacillariophyta was detected belonging of Straminipila and not Oomycota. I strongly suggest the authors to treat only true Fungi and not Straminipila that are evolutionary far from fungi.  In this view, the title also is not correct and must change and considers only true Fungi.

33 “symbiotic fungi. Soils o Clearwater Mesa” change in “symbiotic fungi. Soils of Clearwater Mesa”

Abstract

42 “Basidiobolomycota,” change  punctuation in “Basidiobolomycota;”

Materials and Methods

From the paragraph  2.2. Soil physical and chemical analyses” the line numbers are missing. I hope my corrections are easy to find .

“The granulometric soil characteristics of sand, silt and clay soil chemical analyses”

This phrase is not clear. I think some words are gone. Soil analyses you have carried out are more complex; I suggests the authors to list the analyses that were carried out.

2.3. DNA extraction, Illumina library construction and sequencing

3rd line

“extracted from these using the FastDNA Spin Kit for Soil (MPBIO, Ohio, USA.” Change in “extracted from these using the FastDNA Spin Kit for Soil (MPBIO, Ohio, USA).

Fig. 2

In the figure 2 there are 4 colours and the meaning of one colour (violet) is not reported

Pg 15

“Cultivation study” change in “ a study based on culture-dependent methods “

Author Response

REVIEWER 3

Fungal and fungal-like diversity and ecology assessed using DNA metabarcoding along a deglaciated chronosequence soils at Clearwater Mesa, James Ross Island, Antarctic Peninsula

 The MS in this form need major revision.

Here there are some corrections and considerations

Simple summary

Line 30 “Straminopila” change in “Straminipila” (see Mycobank or other atlas such as Fungal Biodiversity by Crous et al., 2019, Westerdijk Fungal Biodiversity Institute. Change also on line 43 and in all the text, if the authors want to report data on Straminipila. In any case Bacillariophyta contains diatoms and it is better to take it off in the text, all the table, figures and supplementary materials. Moreover it seems that only Bacillariophyta was detected belonging of Straminipila and not Oomycota. I strongly suggest the authors to treat only true Fungi and not Straminipila that are evolutionary far from fungi.  In this view, the title also is not correct and must change and considers only true Fungi.

Answer: we corrected the manuscript with the correct name Straminipila requested by the Reviewer. We also understand the Reviewer concern about our aim to study Fungi and Straminipila kingdom; however, if the Reviewer allows, we would like to keep this fungal-like group in the manuscript, which is poorly known in Antarctica and evolutionary close to Fungi. Keeping Straminipila data in our manuscript can highlight more information about this neglected groupu Antarctica.

33 “symbiotic fungi. Soils o Clearwater Mesa” change in “symbiotic fungi. Soils of Clearwater Mesa”

Answer: the phrase was corrected as requested by the Reviewer.

Abstract

42 “Basidiobolomycota,” change  punctuation in “Basidiobolomycota;”

Answer: the punctuation was corrected as requested by the Reviewer.

Materials and Methods

From the paragraph  “2.2. Soil physical and chemical analyses” the line numbers are missing. I hope my corrections are easy to find .

“The granulometric soil characteristics of sand, silt and clay soil chemical analyses”

This phrase is not clear. I think some words are gone. Soil analyses you have carried out are more complex; I suggests the authors to list the analyses that were carried out.

Answer: the phrase was corrected. I’m sorry about the lines missing; I included the lines in the revised version.

2.3. DNA extraction, Illumina library construction and sequencing

3rd line “extracted from these using the FastDNA Spin Kit for Soil (MPBIO, Ohio, USA.” Change in “extracted from these using the FastDNA Spin Kit for Soil (MPBIO, Ohio, USA).

Answer: the parenthesis was inserted.

Fig. 2

In the figure 2 there are 4 colours and the meaning of one colour (violet) is not reported

Answer: indeed, the violet colour refers to Site 4

Pg 15 “Cultivation study” change in “a study based on culture-dependent methods”.

Answer: the phrase was corrected as requested by the Reviewer.

Best Regards

Luiz Rosa

Reviewer 2 Report

The manuscript by Gonçalves et al. describes the fungal community, as determined by culture-independent approach, in soils collected along a deglaciated chronosequence from para- to periglacial conditions at James Ross Island, Antarctic Peninsula.

The manuscript is well written and I have only minor suggestions:

Please, improve the quality of Figure 1b.

p7, l8, (MPBIO, Ohio, USA)

p8, l21,  ....community. Relative abundance values...

p9, l20-22, ...and might represent taxa not currently included in the available sequence databases or be new Antarctic records and/or previously undescribed fungi". This is a comment. Please, move to discussion section.

p13, l10-12, ... "suggested..., as in the above comment.

p15,  [10,42,43]permafrost, add a space

Author Response

REVIEWER 1

Comments and Suggestions for Authors

The manuscript by Gonçalves et al. describes the fungal community, as determined by culture-independent approach, in soils collected along a deglaciated chronosequence from para- to periglacial conditions at James Ross Island, Antarctic Peninsula.

The manuscript is well written and I have only minor suggestions:

Please, improve the quality of Figure 1b.

Answer: Figure 1 was corrected

p7, l8, (MPBIO, Ohio, USA)

Answer: the parenthesis was included as requested by the Reviewer.

p8, l21, ....community. Relative abundance values...

Answer: the word was corrected as requested by the Reviewer.

p9, l20-22, ...and might represent taxa not currently included in the available sequence databases or be new Antarctic records and/or previously undescribed fungi". This is a comment. Please, move to discussion section.

Answer: the sentence was changed from Results to Discussion section as requested by the Reviewer.

p13, l10-12, ... "suggested..., as in the above comment.

Answer: the sentence was changed from Results to Discussion section as requested by the Reviewer.

p15, [10,42,43]permafrost, add a space

Answer: the space was added as requested by the Reviewer.

Best Regards

Luiz Rosa

Reviewer 3 Report

Dear Authors,

The manuscript investigated the fungal and fungal-like diversity in soils sampled along the deglaciated parts of Clearwater Mesa, James Ross Island, Antarctic Peninsula.

The topic is relevant and interesting. The results and methods are proper.

 The discussion should be more focused and compact. Do not list who found what, rather write e.g. we found this and this species, similarly to X results obtained in Antarctica or lakes etc. 

Question 1: Did you deposit somewhere the sequences which can not be assigned to known phyla or species? They can be useful later for other researchers.

Question 2: 3.1 chapter: You wrote that “A total of 95 fungal ASVs (39.7%) could only be assigned…” Please, clarify whether 88 or 95 ASVs were found.  How many ASV was the total, how many could be determined, and how many remained unknown?

Question 3: In the sample 2 was the highest number of DNA reads and ASVs (Table S4).  Is the site 2 further away from the edges, (from the ice) than the other sites? Is it possible that the microclimate is “warmer” than closer to the ice and edges?

 I have found many inaccuracies, please correct them.

33. line: in …Soils o Clearwater Mesa…     

Did you mean to write that the soils of Clearwater Mesa?

60. line: These ecosystems and the biodiversity they contain face multiple extremes…

Did you mean to write that  These ecosystems and their biodiversity are exposed to multiple extreme conditions???

56. line: I suggest adding to the keywords: metabarcoding

62. line: … experience large and rapid variation…

Did you mean to write that Antarctic soils are dry, and are exposed to large and rapid changes in temperature, and high levels of solar radiation?

82. line: “Antarctic soils is likely to be considerably greater than previously appreciated”.  Greater in size? Richer in microbes? This sentence should be clarified.

correct please, …soils are…

97. line: “Sampling locations were selected to ensure that lichens and mosses were absent”.

Did you mean to write that Sampling locations were selected to be free of lichens and mosses?

Figure 1. The letter c is missing from the photo.

Please clarify the sentence” The distance among the sites sampled”. e.g. b is the distance between site1 and site2?

2.3 chapter: (MPBIO, Ohio, USA             The second bracket is missing.

2.4. chapter:  For fungal classification, we followed and the references …

The and is superfluous.

2.5 chapter: relative abundance values were……”  This sentence should start with a capital letter.

Figure 2: Please specify the legend and captions. Indicates, please what 29.5 and 59 mean on the Y axis. The number of sequence variants? Indicate please, what the numbers in the columns mean.

“Size of each list”   It is not clear.

15. site 22 line? Insert please a space before the permafrost word.

Author Response

REVIEWER 2

Comments and Suggestions for Authors

Dear Authors,

The manuscript investigated the fungal and fungal-like diversity in soils sampled along the deglaciated parts of Clearwater Mesa, James Ross Island, Antarctic Peninsula.

The topic is relevant and interesting. The results and methods are proper.

The discussion should be more focused and compact. Do not list who found what, rather write e.g. we found this and this species, similarly to X results obtained in Antarctica or lakes etc.

Question 1: Did you deposit somewhere the sequences which can not be assigned to known phyla or species? They can be useful later for other researchers.

Answer: Indeed, we did not deposit the sequences. However, as requested by the Reviewer, now they were deposited in GenBank and the codes inserted in the manuscript.

Question 2: 3.1 chapter: You wrote that “A total of 95 fungal ASVs (39.7%) could only be assigned…” Please, clarify whether 88 or 95 ASVs were found. How many ASV was the total, how many could be determined, and how many remained unknown?

Answer: The Reviewer is correct. Indeed, we identified 88 fungal ASVs, among them only 35 (39.7%) were identified in high hierarchical levels. We correct this information in the manuscript.

Question 3: In the sample 2 was the highest number of DNA reads and ASVs (Table S4).  Is the site 2 further away from the edges, (from the ice) than the other sites? Is it possible that the microclimate is “warmer” than closer to the ice and edges?

Answer: This is possible but we do not evidence to support it. We did not detect any physicochemical characteristic that can explain the high DNA reads at site 2.

I have found many inaccuracies, please correct them.

  1. line: in …Soils o Clearwater Mesa… Did you mean to write that the soils of Clearwater Mesa?

Answer: the phrase was corrected.

  1. line: These ecosystems and the biodiversity they contain face multiple extremes…

Did you mean to write that  These ecosystems and their biodiversity are exposed to multiple extreme conditions???

Answer: the phrase was corrected as requested by the Reviewer.

  1. line: I suggest adding to the keywords: metabarcoding

Answer: the word was included as keywords as requested by the Reviewer.

  1. line: … experience large and rapid variation…

Did you mean to write that Antarctic soils are dry, and are exposed to large and rapid changes in temperature, and high levels of solar radiation?

Answer: the phrase was corrected as requested by the Reviewer.

  1. line: “Antarctic soils is likely to be considerably greater than previously appreciated”. Greater in size? Richer in microbes? This sentence should be clarified.

correct please, …soils are…

Answer: Greater in fungal diversity. However, ‘is’ is correct as it refers to ‘fungal diversity’ in the sentence.

  1. line: “Sampling locations were selected to ensure that lichens and mosses were absent”.

Did you mean to write that Sampling locations were selected to be free of lichens and mosses?

Answer: the phrase was corrected.

Figure 1. The letter c is missing from the photo.

Answer: Figure 1 was corrected.

Please clarify the sentence” The distance among the sites sampled”. e.g. b is the distance between site1 and site2?

Answer: The phrase was corrected.

2.3 chapter: (MPBIO, Ohio, USA             The second bracket is missing.

Answer: the parenthesis was included as requested by the Reviewer.

2.4. chapter:  For fungal classification, we followed and the references … The and is superfluous.

Answer: The phrase was corrected.

2.5 chapter: relative abundance values were……”  This sentence should start with a capital letter.

Answer: The sentence was corrected.

Figure 2: Please specify the legend and captions. Indicates, please what 29.5 and 59 mean on the Y axis. The number of sequence variants? Indicate please, what the numbers in the columns mean.

“Size of each list”   It is not clear.

Answer: The Figure 2 was corrected as requested by the Reviewer.

  1. site 22 line? Insert please a space before the permafrost word.

Answer: The space was inserted as requested by the Reviewer.

Best Regards

Luiz Rosa

Round 2

Reviewer 1 Report

Although the ms reports interesting data on fungal biodiversity and ecology of Clearwater Mesa in Antarctic Peninsula that are worth to be published, it cannot be published in this form.  

Unfortunately I can see again the same mistake that I detected in the first version (e.g. in the abstract, line 30 and in the discussion (line 252-253). I don’t agree with the thought of the authors which consider Bacillariophyta as fungal-like organisms. If Straminipila contains  some groups of fungal-like organisms  it doesn’t mean that Bacillariophyta are fungal-like organisms.  I noticed that this mistake is reported in other papers of the same authors. As in the first revision, I suggest the authors to treat only true Fungi and, eventually, discuss the presence of Bacillariophyta as a further result of the method used but which has nothing to do with fungal-like organisms.  

Author Response

Reviewer 1

Although the ms reports interesting data on fungal biodiversity and ecology of Clearwater Mesa in Antarctic Peninsula that are worth to be published, it cannot be published in this form.

Unfortunately I can see again the same mistake that I detected in the first version (e.g. in the abstract, line 30 and in the discussion (line 252-253). I don’t agree with the thought of the authors which consider Bacillariophyta as fungal-like organisms. If Straminipila contains some groups of fungal-like organisms  it doesn’t mean that Bacillariophyta are fungal-like organisms.  I noticed that this mistake is reported in other papers of the same authors. As in the first revision, I suggest the authors to treat only true Fungi and, eventually, discuss the presence of Bacillariophyta as a further result of the method used but which has nothing to do with fungal-like organisms.

Answer: We include Bacillariophyta due the Blastn analysis, which recognize it as Straminipila. However, we understand the concern and we removed Bacillariophyta results from the manuscript as requested by the Reviewer.

Best Regards

Dr. Luiz H. Rosa

Department of Microbiology

Federal University of Minas Gerais

Brazil
